# Shannon Entropy and Informational Redundancy in Minimally Monophyletic Bryophyte Genera

**DOI:** 10.3390/plants14193066

**Published:** 2025-10-04

**Authors:** Richard H. Zander

**Affiliations:** Missouri Botanical Garden, 4344 Shaw Blvd., St. Louis, MO 63110, USA; rzander@mobot.org

**Keywords:** bryophytes, evolution, information, minimally monophyletic groups, redundancy, Shannon entropy, survival

## Abstract

The degree of informational redundancy is often examined in genetic studies but not yet detailed for taxa conceived as minimally monophyletic groups (microgenus). Evolutionary processes in microgenera were reviewed, detailing critical sets of traits, the novon, the immediate ancestron, and the ancestron. Calculations were made from known intra-genus character state changes for maximum entropy, Shannon entropy, and entropic redundancy. Additional evaluations of contrived data sets were intended to evaluate the range of informational variation in small, medium, and large numbers of species and traits. Results indicate that measures of Shannon information and redundancy are rather similar in all but microgenera with the smallest number of species and traits per species. Hypothetically, this similarity is due to the fairly constant balance between numbers of newly evolved traits and traits monothetically redundant because all are shared with all species in the genus. This balance may be explained by a selective construct or emergent property that balances innovation leading to the colonization of new niches and conservation of proven ancestral traits for survival sympatricially and peripatrically in the particular challenges of the ancestor’s niche. The entropic redundancy calculations indicate that 0.20 to 0.30 of the information in a microgenus serves as flexibility in survival adaptation at the genus level.

## 1. Introduction

Computerized methods of evaluating evolutionary relationships are many (review [1]). The use of information theory in systematics has a long history (review [2]), with particular applications in regard to the demand for statistical evaluation of molecular phylogenetics and in genomics. Claude Shannon’s [3] use of the thermodynamic term “entropy” in the analysis of information quality has been a conceptual sticking point, but the protocols are clear for evaluating the degrees of certainty and uncertainty in data. Shannon entropy, also known as Shannon diversity, is a measure of the uncertainty in information content of a message. Here, it describes the amount of apparently uninformative baggage in the trait set (a trait is a character state) of a genus. That “baggage” turns out to be valuable in survival of the genus [4].

Actual known values for Shannon diversity are generally related to biological diversity of certain ecosystems. For instance, species richness in coral reefs [5] was compared across transects with Shannon entropy measures ranging from 1.1 to 3.2 across transects, averaging 2.2, while in sponge biome composition Shannon entropy varied from 0 to 8.1, averaging 3.4. The authors report that, most simply, coral communities are on average less diverse than sponge microbiome communities because they have a significantly lower Shannon entropy. The authors, however, warn that this is biased by the large difference in taxon richness, and the consideration of upper and lower bounds of index values actually reverses this comparison.

There has been much published on the information content of cladograms, for instance, the length of branches in molecular phylograms, but little on morphological trait redundancy. There is little actual data detailing the relative information content within and between genera other than distance measures between inferred hypothetical common ancestors.

The present study used data from previously revised sets of species [6,7], which allowed the calculation of maximum entropy and Shannon entropy, and from these, entropic redundancy using standard formulae. What is novel about this study is the framework of structural monophyly allowing these theoretical information calculations for an informationally little-studied group, the bryophytes. Structural monophyly proposes that taxa larger than about eight species are probably made up of smaller, “microgenera,” that consist of one ancestral species and a few, usually averaging about four, descendant species. The microgenus is a minimally monophyletic group (MMG) and is nomenclaturally treated as a genus.

The species of a microgenus are monothetic and evolutionarily connected because they each share the newest traits (averaging about four) of the ancestral species. The newest traits are a set called the “novon.” The value of redundancy in evolution is unquestionable, and the present paper is a first effort at measuring the redundancy of the set of shared traits in a microgenus. It is projected here that, while the species is the basic unit of evolution, the MMG is basic causal framework. The MMG is the factory where the elements of evolution are crafted, balanced, and pruned by natural selection while enhancing biological diversity.

### 1.1. Past Applications of Shannon Entropy in Evolution

Measures of relative richness of biological diversity are commonly limited to numbers and rarity of species, but may be enhanced with information on functional diversity and functional redundancy [8,9]. It has been established that the functional redundancy associated with high species diversity provides for more stable host–microbiome interactions.

Calculation methods and measures of functional redundancy are diverse and difficult to compare across biological groups [10,11,12,13]. Functional redundancy at the ecosystem level—that of a complex adaptive system—is hypothesized to increase ecological resilience and stability, because buffering ecosystem community functions with more redundant species (with similar functions) mitigate against the loss of individual species by ecosystem disturbance [14,15,16]. Stochastic processes alone help preserve species richness due to functional redundancy [13,14].

There is significant literature on functional redundancy and its contribution to adaptation (e.g., [17]), but little work has been conducted in the context of evolutionary systematics. Most analyses are at the gene or species level, with an emphasis on ecology, and little comparison is made between genera or higher taxa. In the literature, redundancy is commonly associated with paralogy, the presence of duplicated genes in the genome. These paralogous genes may increase certain physiological outputs, but more importantly undergo over time, neofunctionalization, which is the process of mutations changing function as a retreat from redundancy [18]. Few studies directly report redundancy measures calculated from Shannon entropy.

### 1.2. The Minimally Monophyletic Group

The MMG is illustrated by a caulogram showing inferred ancestral connections between species (Figure 1). Van Valen [19] early argued against cladists that extant speciesthat were clearly direct ancestors to other extant species are actually common. I have found that about half of the species studied [6,7] are immediate ancestors of other species. The MMG defines the taxonomic group on a step higher than species, the genus, of commonly of about five species, one of which is progenitor, all of which share the newly evolved traits of the progenitor (the “immediate ancestron”). The descendant species generate their own new traits, about four new traits in their own novons, from state changes in older, less survival-important characters.

Deciding which species of a small group are MMGs helps decide how to break up a larger, polythetic group. Three criteria are involved. Species that do not fit into a group by being distinguished from the remainder by having more than about eight traits difference, are segregated into a different microgenus.

Of the set of most-similar species, one of them is tagged as the progenitor by two criteria (second-order Markov chain): of all species it is the one most similar to an outgroup, and it is the most generalist (least specialized) among the ingroup [6], p. 8. Given the initial small size of a microgenus, it is easy to test relationships for parsimony in creating a model of an ancestor-generating descendants by second-order Markov criteria. By thus minimizing or, better, avoiding reversals, the problem may be treated much like a simplex system.

Microgenera (an MMG) are well-supported statistically by Shannon–Turing sequential Bayesian analysis [6], p. 9. The newly evolved traits of each species are considered actors in evolution for their central role in speciation, and as adaptively contributing to stasis over millions of years. If each different trait is assigned one informational bit (the minimum certainty needed to make a decision), then computation is easier (bits are logarithmic and can be added). Turing sequential Bayesian analysis, when used, generates high support by Bayesian posterior probabilities for most analyzed microgenera.

Given that structural monophyly is perhaps unfamiliar to practicing cladists, an illustrative diagram (Figure 1) shows the essentials. The plenum in which the process is played out is the “ancestron,” being the total set of character trait changes available within a genus, tribe, or family. The central operator in the speciation/evolution context is the minimally monophyletic genus (MMG), labeled “genus ancestor” in Figure 1, where it looks like a baby octopus. The newly evolved (mutation plus fixation) trait set is called the “novon.”

The microgenus’ basic components are those involved with sorting traits that contribute to evolutionary differences between species. The genus ancestor receives (powered by natural selection) the newest traits of its own ancestor, which is now called the “immediate ancestron.” The ancestor’s ancestor is a descendant species itself, and the genus ancestor mimics the same evolutionary processes (probably also through natural selection) as its own ancestor.

Descendant species, graphically depicted as baby octopus’ tentacles in Figure 1, are optimally four in number. The genus ancestor generates descendant species theoretically [6], p. 9, rather abruptly during a geological period of about 22 million years. Descendant species then generate new descendants. Thereafter, species, and eventually genera, go extinct over a period of more than 100 million years for the taxonomic group studied in this paper. During the survival time of each species, their “immediate ancestron,” which duplicates the ancestor’s novon, is paired with a set of newly evolved traits (stochastically generated, trimmed in number to only a few), and fixed by success in relative isolation. The irregular bottom of Figure 1 represents, graphically, the interplay of environment and organism in adaptation and mutualism.

That there are optimally four species per genus in minimally monophyletic groups (at least of the 23 groups studied to date) is reflected in the fact that in a Gaussian distribution, 0.95 of uncertainty is eliminated in a standard deviation of 2, which is met when the novon is of four or five traits, where one trait is equivalent to one Shannon information bit. Two-sigma structural monophyly will be examined more closely in another paper.

## 2. Results

The analysis (Table 1) of five microgenera exemplifies the protocol of using somewhat morphologically disparate taxa. A formula is given with letters, separated into species, representing novel traits of descendants; for instance, “ab” represents the number of traits (2) in the novon of *Ardeuma recurvirostrum*). Numerals in the formula represent redundant, shared traits in both the progenitor and all descendants; for instance, “1234” repeated for each of the five species in *Ardeuma.* The whole formula is analyzed for Shannon entropy, with spaces ignored. The caulograms (charted evolutionary tree in spreadsheet format), although part of larger previously published caulograms [4,6,7], are of microgenera (MMG) here treated as standalones. What is immediately clear is that maximum entropy, Shannon entropy, and entropic redundancy are nearly the same in all MMGs, even with different microgenera of different sizes. This is apparently due to the similar, balanced numbers of fully unique traits and of fully redundant traits. Apparently both sorts of traits are important in speciation and subsequent survival.

The microgenera *Ardeuma,* of five species, *Gymnostomum,* of four species, and *Hymenostylium,* of two species are in the Pottiaceae tribe Pleuroweisieae, characteristic of hygric habitats worldwide, like the ambience around waterfalls. *Stephanoleptodontium* is the largest genus in the related moss family, Streptotrichaceae, characteristic of arboreal and soil sites in mountainous areas of the tropics and subtropics, while *Tainoa* is in Pottiaceae tribe Trichostomeae, this genus restricted to tropical and subtropical soil habitats around the Caribbean. The environments are, to a large extent different, for each microgenus.

All these MMGs have about the same critical measures of informational entropy and redundancy. Maximum entropy ranges from 3.58 to 5.64, increasing somewhat with the size of the group. Shannon entropy ranges from 2.75 to 4.13, again increasing with the size of the group. Informational redundancy ranges from 0.23 to 0.31, not well correlated with group size, but apparently affected by the relatively large number of new traits of the progenitor.

The high redundancy of *Gymnostomum* is due to the five newest traits of the progenitor shared by all descendants, which are accumulated among the four descendant species (while the number of new traits in each species are typically not so high. *Hymenostyliella,* of two species, has low redundancy even with five traits in the genus progenitor’s novon. The small size of the genus, apparently affects the probabilistic results of entropy calculations, as demonstrated in Table 2, the MMG of two species, two to six traits. It has been pointed out that small samples may bias analysis [9,20].

Table 2 demonstrates that the similarity of analytic results in actual microgenera is replicated in contrived analyses intended to examine the extreme stuffing of data Excepting lower values for redundancy in the MMG of two species, data sets generated about equal R values between 0.18 and 0.34 with most values between 0.26 and 0.21. Although about half of the speciationally important traits in the average microgenus are redundant, in information theory the entropic redundancy value is about half that.

An explanation of low values for the redundancy is not intuitively clear due to the probabilistic bent of Shannon Entropy. Traits that are redundant are probabilistically more predictable, where maximum entropy is most predictable. A redundant trait carries less information than a unique trait. A redundancy of 0.20, as is common in the present data set, means that the observed data contains 0.20 less information than the data could, if all symbols (novel traits) appeared with equal probability. It indicates that 0.20 of the potential for new (surprising) information is information space taken up by predictable patterns, e.g., here, the repeated trait sequence “12345.”

Data with a high redundancy have low Shannon entropies, for instance, “12345 12345” has a Shannon entropy of 2.75. The much higher Shannon entropies of the data from actual species reflect the information-rich data from unique traits (novons) and is not too affected by the strong complement of redundant data. One might suggest that this may be correlated with the struggle for immediate survival upon speciation and long-term survival though environmental vicissitudes where redundancy comes into play.

## 3. Discussion

In the above analysis, every minimally monophyletic genus has about the same level of informational redundancy in the traits considered most important for survival and speciation, that is, the newest traits fixed at speciation. Because this feature is present in differently sized genera in three different large groups of mosses, it should be considered an emergent phenomenon generated by the differential survival of organisms at the genus level. The manner that redundancy of traits is passed along a caulogram (stem-taxon evolutionary tree) is diagrammatically presented in Figure 2. Each ellipse is a species with newest traits in the terminal half of the ellipse. The basal half of the ellipse represents the species contribution from the newest traits of its own progenitor. A progenitor’s newest traits are shared among all immediate descendants, represented by spreading of color into the descendant ellipse.

The number of critical new traits associated with speciation in this group is generally about four, of course limited to taxonomically identifiable morphology. There may be different expressed molecular (exon), physiological, or ecological traits, perhaps limited to the same small number by whatever is governing the morphological traits. This implication should be investigated.

A caulogram (Figure 2B) integrates the series (Figure 2A) more clearly as a dendrogram. Black filling simply represents other traits generated and fixed in other descendant species, but the focus is solely on the series of three microgenera connected by their progenitors. This model is the process-based context of the data presented in Table 1 and clarified in Table 2.

The parallel to redundancy at the gene level is that redundancy at the genus level may act in the same manner to buffer against threats to survival, providing informational flexibility at the genus level. In every genus, a copy of the traits of the progenitor in each descendant is a passport to the goal of an appropriate peripatric or allopatric niche and serves to coddle the character-state changes in unused traits that will constitute the new descendant’s adaptation to a new niche or just temporary stasis in mutualism.

A microgenus with maximum entropy would have no species with new traits redundant by matching. I suggest that 0.20–030 of the information content in a microgenus is dedicated to gifting descendant species with a recently proven set of highly survivable traits in the sympatric and peripatric areas of the progenitor’s ecological niche. This kind of altruism-at-a-distance makes sense in the context of natural selection for survival at the genus level and perhaps at higher taxonomic levels. Such traits might be of survival value as niches are expanded and contracted over time by climate change and geological events that require the co-existence of sibling descendant species. The survival of flexible, otherwise well-adapted, well-integrated genera must add to the survival of an entire ecosystem.

The use of Shannon entropy with language-mediated messages involves the calculations of the different probabilities of their occurrence in real information transfer. In the case of this analysis, the symbols used as proxies for character states are assigned probabilities according to their appearance in the symbol string (the message) ignoring spaces. Of course, it might be worthwhile to chart the number of occurrences of each trait in a taxonomic revision as a parallel to alphabet occurrence in real life, but it may be stated that examination of actual MMGs finds that all new traits are unique in the microgenus, and all redundant traits are easily assigned, by software, to their actual probabilities *for that microgenus*.

## 4. Materials and Methods

Analysis was made of maximum entropy, Shannon entropy, and entropic redundancy, for example, microgenera, both actual and contrived. Data for the analysis (Table 1) of three moss genera, *Ardeuma, Gymnostomum,* and *Hymenostyliella*, are from an analysis of MMGs in the moss tribe Pleuroweisieae [7] of the moss family Pottiaceae, for the microgenus *Stephanoleptodontium* from a revision of the related moss family Streptotrichaceae [6], p. 4, and for *Tainoa* from a study of Pottiaceae subfam. Trichostomoideae in the West Indies [8]. Data are formatted for presentation to software analysis by PlanetCalc’s Shannon entropy calculator [21], which is the only calculator now available online that can be set to ignore spaces in symbol strings. Data were also invented for this study to cover common scenarios in MMGs for the analysis of variation in extreme but possible structural monophyly.

For Table 1 of actual microgenera, the number of traits important in speciation for each species is as read off a published caulogram [6,7,8]. Included data for each microgenus are the novons of each of the species, including the novon of the progenitor but not its immediate ancestron (which are traits that belong to its own progenitor’s microgenus). These traits are treated as initially equally probable. The actual probability of each trait in the symbol string (the message) is the number of instances in the symbol string divided by the total number of symbols in the string. Maximum entropy is simply the logarithm to the base 2 of that total number of all symbols (*m* in Equation (1)) in the string (Equation (1)). *H_max_* is the maximum possible entropy for a system with a given number of possible states or symbols, all states equiprobable.(1)$$H_{max}={log}_{2}\left(m\right)$$

The Shannon entropy value is then calculated with the PlanetCalc online module following the standard formula (Equation (2)), given the actual probabilities of each symbol appearing in the symbol string (number of appearances divided by total number of symbols).(2)$$H\left(X\right)=-\sum P\left(x\right){log}_{2}P\left(x\right)$$

The Shannon entropy value is then subtracted from the maximum entropy to obtain a differential value for redundancy, with is normalized by dividing by maximum entropy (Equation (3)). The result is *R,* which is entropic redundancy. In information theory, redundancy is defined as the difference between the maximum possible entropy of a source and its actual entropy. It is a measure of how much informational space is less informative than it could be. Relative redundancy is calculated and is given in Table 1. The inclusion of *H_max_* in the denominator normalizes the measurement to allow for comparisons.(3)$$R=\frac{H_{max}-H(X)}{H_{max}}$$

For the contrived data of Table 2, data were created to stand for various numbers of species in a microgenus with various numbers of traits in each species. In this case, two species with two to six traits in each, three species with two to six traits, and the same for four and five species. This covers much of the expected variation in nature. One unique alphanumeric symbol stands for each unique trait. Letters of the alphabet are used for traits of the novons of descendants while numerals stand for the novon of the progenitor, which becomes redundant as it occurs in all descendants.

There is a space separating the symbols for each species represented, all species have the redundant traits given (duplicated numeral sets), while one less species is represented by letters because only the novon traits of the progenitor are listed, the traits given to it by its outgroup ancestor are not considered informative of redundancy in the microgenus. For instance, in the MMG of five species (Table 2), ab cd ef gh 12 12 12 12 12 provides letters for four species (novons of descendants) and numerals for five species (shared, redundant traits of the progenitor and all descendants). The actual probability of the unique symbol a is 1/18 and that of the redundant symbol 1 is 5/18. Each different symbol stands for a different trait.

## Figures and Tables

**Figure 1 plants-14-03066-f001:**
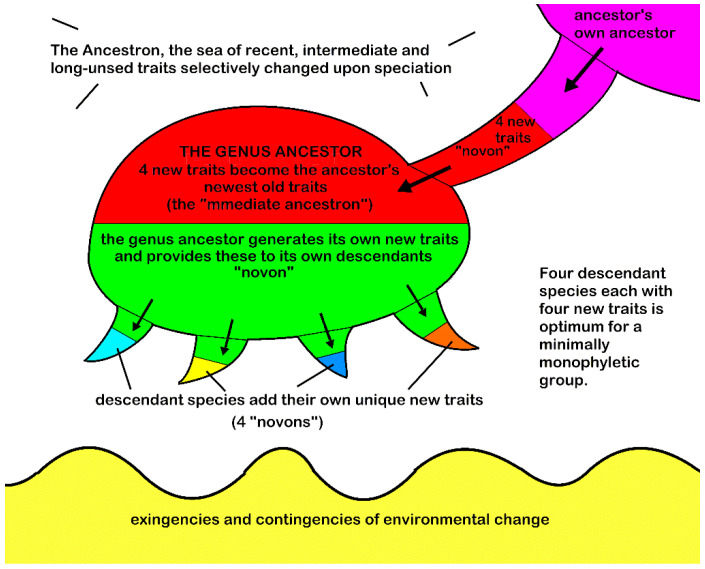
Diagrammatic representation of speciation processes in a microgenus. The progenitor has advanced traits (red) from its own ancestor (purple) and is characterized by its own newly evolved traits (green). Its descendant species (usually four) possess the progenitor’s new traits (green) and evolve their own new traits (azure, yellow, blue, orange) that are adaptations to variations in the environment (wavy yellow substrate). The background (white) is the set of all characters available to species in the larger group (tribe, subfamily, family) for which each state is a potential trait.

**Figure 2 plants-14-03066-f002:**
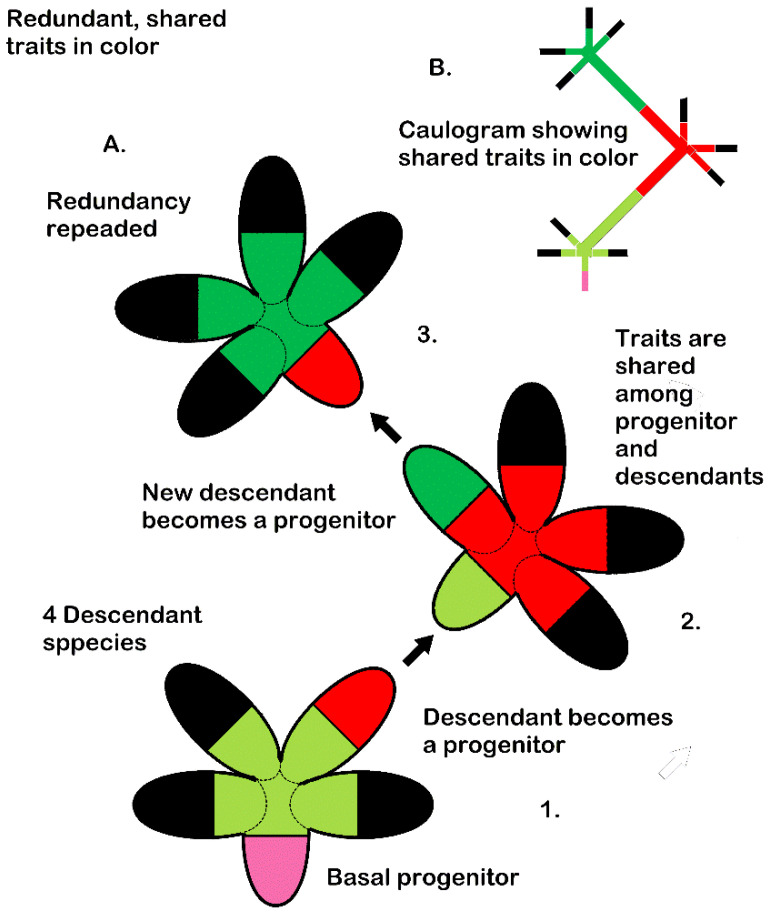
Model of redundancy across an evolutionary tree. Each ellipse represents a species with the progenitor’s newest traits (terminal portion of ellipse) shared among all (optimally four) descendant species. (**A**). Detail of transformation of (1) a descendant species into a progenitor of another microgenus (2), and then again with a later descendant (3), sharing transferred newest traits (same color). (**B**). A caulogram summarizing transformations involving redundancy through an evolutionary series of microgenera.

**Table 1 plants-14-03066-t001:** Minimally monophyletic genera in Pottiaceae tribe Pleuroweisieae. Evolutionary tree (caulogram) in spreadsheet implementation, total traits important in speciation, and maximum entropy, Shannon entropy, and entropic redundancy are given.

Genus	Species	New Taits (the Novon Set)			Total Evol. Traits	Max. Entropy, Shannon Entropy, Redundancy
*Ardeuma*	*gracillimum*	4			4	
	*recurvirostrum*	╚>	2		4 + 2	
	*crassinervium*		╠>	3	4 + 3	
	*annotinum*		╠>	3	4 + 3	
	*aurantiacum*		╚>	3	4 + 3	
Total					31	4.95, 3.72, 0.25
Formula: ab cde fgh ijk 1234 1234 1234 1234 1234						

*Gymnostomum*	*aeruginosum*	5			5	
	*viridulum*	╠>	2		5 + 2	
	*calcareum*	╚>	3		5 + 3	
	*mosis*		╚>	2	5 + 2	
Total					27	4.75, 3.27, 0.31
Formula: ab cde fg 12345 12345 12345 12345						

*Hymenostyliella*	*llanosii*	5			5	
	*alata*	╚>	2		5 + 2	
Total					12	3.58, 2.75, 0.23
Formula: ab 12345 12345						

*Stephanoleptodontium*	*longicaule*	4			4	
	*syntrichioides*	╠>	4		4 + 4	
	*brachyphyllum*	╠>	3		4 + 3	
	*filicola*	║	╚>	5	4 + 5	
	*capituligerum*	╚>	3		4 + 3	
	*latifolium*		╠>	3	4 + 3	
	*stoloniferum*		╚>	4	4 + 4	
Total					50	5.64, 4.13, 0.27
Formula: abcd efg hijkl mno pqr stuv 1234 1234 1234 1234 1234 1234; 1234						

*Tainoa*	*pygmaea*	4			4	
	*sinaloensis*	╠>	5		4 + 5	
	*subangustifolia*	╠>	3		4 + 3	
	*subcucullata*	╚>	4		4 + 4	
	*bartramiana*		╚>	3	4 + 3	
Total					35	
Formula: abcde fgh ijkl mno 1234 1234 1234 1234 1234						5.13, 3.80, 0.26

**Table 2 plants-14-03066-t002:** Contrived data sets modeling microgenera with different sets of novons (letters) and redundant traits shared by the progenitor (numeric), organized in sets for different numbers of species and traits. Total traits are used to calculate maximum entropy, Shannon entropy is calculated with online software, and entropic redundancy follows the standard formula.

Formulae	Total Traits	Max Entropy	Shannon Entropy	Entropic Redundancy
**MMG of 2 species, 2 to 6 traits**				
ab 12 12	6	2.58	2.24	0.13
abc 123 123	9	3.17	2.72	0.14
abcd 1234 1234	12	3.58	3.09	0.14
abcde 12345 12345	15	3.91	3.38	0.13
abcdef 123456 123456	18	4.17	3.62	0.13
**MMG of 3 species, 2 to 6 traits**				
ab ce 12 12 12	10	3.32	2.59	0.22
abc def 123 123 123	15	3.91	3.11	0.20
abcd edgh 1234 1234 1234	20	4.32	3.39	0.22
abcde fghij 12345 12345 12345	25	4.64	3.79	0.18
abcdef ghijkl 123456 123456 123456	30	4.91	4.03	0.18
**MMG of 4 species, 2 to 6 traits**				
ab ce ef 12 12 12 12	14	3.81	2.71	0.29
abc def ghi 123 123 123 123	21	4.39	3.37	0.23
abcd edgh ijkl 1234 1234 1234 1234	28	4.81	3.59	0.25
abcde fghij klmno 12345 12345 12345 12345	35	5.13	3.99	0.22
abcdef ghijkl mnopqr 123456 123456 123456 123456	42	5.39	4.25	0.21
**MMG of 5 species, 2 to 6 traits**				
ab cd ef gh 12 12 12 12 12	18	4.17	2.77	0.34
abc def ghi jkl 123 123 123 123 123	27	4.75	3.46	0.27
abcd edgh ijkl mnop 1234 1234 1234 1234 1234	36	5.17	3.82	0.26
abcde fghij klmno pqurs 12345 12345 12345 12345 12345	45	5.49	4.20	0.24
abcdef ghijkl mnopqr stuvwx 123456 123456 123456 123456 123456	54	5.75	4.46	0.23

## Data Availability

All data are available in the publications cited in this paper.

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
