# Peer review of "Shannon Entropy and Informational Redundancy in Minimally Monophyletic Bryophyte Genera"

_plants, 2025, doi:10.3390/plants14193066_

Round 1
Reviewer 1 Report
Comments and Suggestions for Authors
In his manuscript, Richard Zander proposes a highly interesting application of Shannon entropy and redundancy to the study of bryophyte evolution. His views presented here and in some of his prior works are original and intellectually stimulating.
Nevertheless, I would like the author to make some changes and add some more information.
He sometimes makes strong claims about his recent work that lack support from other researchers. For example, in lines 36-37, he writes that that “baggage” turns out to be valuable in survival of the genus or in lines 114-115 when he states that microgenera (a MMG) are well-supported statistically by Shannon-Turing sequen-
tial Bayesian analysis. Perhaps the use of “suggest” or similar can soften the statements.
I have some questions concerning the methods the author uses. Talking about when to separate microgenera, he indicates that species that do not fit into a group by being distinguished from the remainder by having more than about eight traits difference, are segregated into a different microgenus. It seems that this number worked well in the cases he studied and with the data he used, but would this not depend partially on the group studied and the methods used (for example, scanning electron microscopy added)?
Richard Zander also seems to be very strict in the number of traits that define new species and the number of new species that belong to an MMG. I would expect these values to vary depending on different evolutionary scenarios, for example, a widely distributed ancestor with many peripheral populations that become geographically isolated and may or may not be subject to selective pressure. Selection can also be driven by physiological adaptation that is not visible or even molecular changes with no evident selective advantage, but that may impede genetic contact.
For many years, DNA sequence data have become very important in systematics. Consequently, I would appreciate some lines of discussion dedicated to the relation of his morphology-based treatment to molecular data. I can see that a species is morphologically stable after originating one or several descending species. In an initial phase, I would also expect that using neutral molecular markers, we find first a polyphyletic-paraphyletic phase, but over time, this situation should evolve and proceed to monophyly. I think that this fact is somehow related to the caulogram presentation, although it is not equivalent.
Another complication that is not mentioned is the importance of hybridization. At present, there is not too much literature on hybridization in bryophytes in general and Pottiaceae in particular, but there is no doubt that it plays a certain role in the evolution of these organisms. Therefore, the author can perhaps add a line or two to comment on this theme.
There are minor typos that should be corrected:
Line 29: [2])
Line 88: paralogy
Line 96: Van Valen
Line 114: an MMG
Line 256: matching. I suggest [period missing?]
Line 381: https://doi.org/10.15407/ukrbotj81.02.087 [remove :22]
Author Response
Reviewer 1:
In his manuscript, Richard Zander proposes a highly interesting application of Shannon entropy and redundancy to the study of bryophyte evolution. His views presented here and in some of his prior works are original and intellectually stimulating.
Nevertheless, I would like the author to make some changes and add some more information.
He sometimes makes strong claims about his recent work that lack support from other researchers. For example, in lines 36-37, he writes that that “baggage” turns out to be valuable in survival of the genus or in lines 114-115 when he states that microgenera (a MMG) are well-supported statistically by Shannon-Turing sequen11-
tial Bayesian analysis. Perhaps the use of “suggest” or similar can soften the statements.
AUTHOR’S RESPONSE: There ARE no other researchers who are working on structural monophyly in terms of descent with modification. All are cladists using shared ancestry, which cannot resolve the information I am finding. The poaper cited [4] explains that this is not a suggestion but a firmly supported inference.
I have some questions concerning the methods the author uses. Talking about when to separate microgenera, he indicates that species that do not fit into a group by being distinguished from the remainder by having more than about eight traits difference, are segregated into a different microgenus. It seems that this number worked well in the cases he studied and with the data he used, but would this not depend partially on the group studied and the methods used (for example, scanning electron microscopy added)?
AUTHOR’S RESPONSE: Absolutely. It would depend on the groups studied. However, I’ve studied 23 quite different groups and they all segregate in the fashion I report. The traits used are those in the same scale, yes It could be that traits dealing with other physiological and dimensional features may show some different results but the traits I use are directly involved in evolution. The reviewer may wish to use other traits in his/her own studies, and report any conflict.
Richard Zander also seems to be very strict in the number of traits that define new species and the number of new species that belong to an MMG. I would expect these values to vary depending on different evolutionary scenarios, for example, a widely distributed ancestor with many peripheral populations that become geographically isolated and may or may not be subject to selective pressure. Selection can also be driven by physiological adaptation that is not visible or even molecular changes with no evident selective advantage, but that may impede genetic contact.
AUTHOR’S RESPONSE: Yes, I am strict but only in reporting the actual facts, such as those in the multifarious graphs in Willis’ Age and Area book. These hollow curves summarize a vast number of different taxa, all with various selection pressures and histories. I have added a paragraph suggesting a cause for two-sigma structural monophyly.
For many years, DNA sequence data have become very important in systematics. Consequently, I would appreciate some lines of discussion dedicated to the relation of his morphology-based treatment to molecular data. I can see that a species is morphologically stable after originating one or several descending species. In an initial phase, I would also expect that using neutral molecular markers, we find first a polyphyletic-paraphyletic phase, but over time, this situation should evolve and proceed to monophyly. I think that this fact is somehow related to the caulogram presentation, although it is not equivalent.
AUTHOR’S RESPONSE: This is dealt with in my previous paplers, cited in the References. The convolute theory associated with trying to match molecular studies with evolutionary theory is replaced in large part by the simple logic of descent with modification using morphological traits. Given that about half of all extant species are ancestors of at least one other species, the logic of shared ancestry without identifying which species are clearly ancestral and which are descendant is negligent. The valuable part of molecular analysis is the identification of apophyly-paraphyly pairs, which can identify ancestor-descendant relationships; but this is never done. I have published the first instance of this in my “integrative” paper, which the ancestor-descendant molecular pair actually matches the morphological pair. Monophyly is ensured by high Bayesian support for both molecular and morphological studies, and mapping is then not necessary.
Another complication that is not mentioned is the importance of hybridization. At present, there is not too much literature on hybridization in bryophytes in general and Pottiaceae in particular, but there is no doubt that it plays a certain role in the evolution of these organisms. Therefore, the author can perhaps add a line or two to comment on this theme.
AUTHOR’S RESPONSE: I have recognized no evidence of hybridization. Some species are closely related but no evidence of fusion of trait sets is t be found. When a research does find such evidence, then that researcher should discuss it, but that is not presently myself.
There are minor typos that should be corrected:
AUTHOR’S RESPONSE: I thank the reviewer for his/her clear help and concern with the details of this paper. The additional through-provoking criticisms were valuable. All typos are corrected.
Line 29: [2])
Line 88: paralogy
Line 96: Van Valen
Line 114: an MMG
Line 256: matching. I suggest [period missing?]
Line 381: https://doi.org/10.15407/ukrbotj81.02.087 [remove :22]
Reviewer 2 Report
Comments and Suggestions for Authors
This is an interesting manuscript that needs further methodological work:
1) A key factor is making the central claim testable (0.20–0.30 redundancy). Right now the “0.20–0.30” range is presented as a qualitative regularity (Results & Discussion). Report point estimates with uncertainty (e.g., bootstrapped 95% CIs) for each microgenus, then meta-analyze across microgenera (random-effects). This turns an observation into an inference. Indeed, the author should add a sensitivity analysis showing how R changes if (i) you vary the number of “novon” traits per species, (ii) you change the rule “ignore spaces,” and (iii) you reassign borderline traits.
2) Alongside entropic redundancy R=(Hmax−H)/HmaxR, the author should report effective numbers (Hill numbers, like exp Shannon) so readers can interpret results on a linear “diversity” scale.
3) Link “redundant progenitor traits” to ecological buffering and niche tracking in bryophytes (e.g., desiccation tolerance suites, substrate preferences). Consider adding one worked case study (e.g., Gymnostomum) where you narrate which traits plausibly buffer survival and cite habitat/functional literature
4) The author should improve the title to give a more celar emphasis on the results obtained. I suggest: “Shannon entropy and informational redundancy in minimally monophyletic bryophyte genera”
Author Response
Reviewer2:
This is an interesting manuscript that needs further methodological work:
AUTHOR’S RESPONSE: This is a novel and introductory application. Further methodological work must come from future studies by other researchers.
- A key factor is making the central claim testable (0.20–0.30 redundancy). Right now the “0.20–0.30” range is presented as a qualitative regularity (Results & Discussion). Report point estimates with uncertainty (e.g., bootstrapped 95% CIs) for each microgenus, then meta-analyze across microgenera (random-effects). This turns an observation into an inference. Indeed, the author should add a sensitivity analysis showing how R changes if (i) you vary the number of “novon” traits per species, (ii) you change the rule “ignore spaces,” and (iii) you reassign borderline traits.
AUTHOR’S RESPONSE: This is a quantitative regularity based on observations of well-described taxa (many observations) that are ordered in evolutionary series with massive support (95-99%) from Shannon-Turing sequential Bayesian analysis. There is no need to bootstrap CI confidence intervals because uncertainty is already well established as very small. The request for a sensitivity analysis is supererogatory because large numbers of graphs of species per genus in many groups are already published by Willis (Age and Area paper) and others. “Ignore spaces” is nonsense, they are artificial. The ”borderline traits” are dealt with in the taxonomic papers on which the taxa are based.
- Alongside entropic redundancy R=(Hmax−H)/HmaxR, the author should report effective numbers (Hill numbers, like exp Shannon) so readers can interpret results on a linear “diversity” scale.
AUTHOR’S RESPONSE: Hill numbers , species diversity in the community, and dominance is not the focus of this paper. Effectiveness and Shannon diversity is a tangential aspect of redundancy, yes, but is a different study and should be pursued by an ecologist.
- Link “redundant progenitor traits” to ecological buffering and niche tracking in bryophytes (e.g., desiccation tolerance suites, substrate preferences). Consider adding one worked case study (e.g., Gymnostomum) where you narrate which traits plausibly buffer survival and cite habitat/functional literature.
AUTHOR’S RESPONSE: Again, the reviewer asks for an ecological adjunct to my evolutionary study. Certainly such a study would be significant and telling, but it is not the point of the present paper.
- The author should improve the title to give a more celar emphasis on the results obtained. I suggest: “Shannon entropy and informational redundancy in minimally monophyletic bryophyte genera”
AUTHOR’S RESPONSE: Excellent idea! I have changed the title to this reviewer’s suggestion. I also thank this reviewer for his/her incisive and thought-provoking comments even if I’ve discounted their applicability in the present paper.
Round 2
Reviewer 2 Report
Comments and Suggestions for Authors
The author responded adeqautely to my concerns. Now the manuscript can be accepted for publication.